# Flashbulb Memories in the Communication of the Diagnosis of Visual Impairment: The Effect of Context and Content

**DOI:** 10.3390/children10050881

**Published:** 2023-05-14

**Authors:** Mª Emma Mayo Pais, José Eulogio Real Deus, Patricia Mª Iglesias-Souto, Eva Mª Taboada-Ares

**Affiliations:** 1Department of Developmental and Educational Psychology, Faculty of Psychology, University of Santiago de Compostela, 15782 Santiago, Spain; 2Department of Social, Basic and Methodological Psychology, University of Santiago de Compostela, 15782 Santiago, Spain

**Keywords:** visual impairment, diagnosis, communication, first news, flashbulb memories

## Abstract

The diagnosis of a child’s visual impairment is remembered vividly and intensely by their parents. However, the way in which the diagnosis is communicated may affect the development and persistence of this memory. The aim of this study is to analyze the circumstances in which the first news of the diagnosis of visual impairment in children is given and whether the memory of this event persists over time leading to a flashbulb memory. A longitudinal study was carried out with the participation of 38 mothers. Data were collected on sociodemographics, clinical variables, circumstances surrounding the communication of the diagnosis, and the degree of agreement of the information in the two phases of the research. The diagnosis was, on the whole, given to both parents at the same time, in medical language and with little tact, generally in the office of an ophthalmologist. The mothers would have preferred to have received the news in a different way, and the existence of a flashbulb memory is confirmed, more dependent on the context in which the diagnosis was given and its content than on sociodemographic and clinical factors. The way in which the first news of such a diagnosis is given plays a significant role in how it is remembered. Therefore, an improvement in medical practice regarding the communication of such diagnoses is recommended.

## 1. Introduction

Brown and Kulik [1] coined the term flashbulb memory (FBM) to refer to a particularly vivid memory generated in response to a unique, highly emotional experience that becomes etched in the memory like a “flashbulb” or “snapshot” of a moment when everything stopped as if it were “frozen”. Such memories prove impossible to forget and will, therefore, be remembered for life. The basic characteristic of these memories is their emphasis on peripheral aspects such as where the person was when they heard the news, what they were doing, how they heard the news, the emotional responses of those present, and the emotions felt, among other aspects [2]. The vividness and clarity of these memories is so evident [1] that, initially, it was believed that there was a neurophysiological mechanism responsible for remembering the information in such a detailed manner [3]. Based on this belief, Brown and Kulik [1] put forward the first model to explain the formation of FBMs, with another three models subsequently being formulated [4,5,6]. All four models coincide in that, for an FBM to be formed, cognitive, emotional, and social aspects must be involved.

In accordance with the first model [1], known these days as the photographic model, the two main factors that contribute to the formation of FBMs are the reaction of surprise when hearing the news for the first time and the evaluation of the importance/consequence thereof [7]. Brown and Kulik [1] suggested that the surprise generated by an unexpected event precedes the evaluation of that event in terms of its importance/consequence and leads to a high degree of emotional activation (arousal) [8]. They also maintained that the importance/consequence attributed to the event, along with an intense emotional activation, stimulated greater conversation about it [8]. The second model [4], known as the comprehensive model, suggests that two latent variables (emotionality and retelling or repetition) could have direct effects on the formation of FBMs and that indirect effects could be caused by knowledge/interest or importance/consequence.

The third model, put forward by Finkenauer et al. [6], suggests that the process of the formation and retention of FBMs could arise in two ways. The first way is direct, going from the shock to the cognitive evaluation of the news and thus to the FBM, whereas the second is indirect and highlights the fact that the formation of the FBM begins with the cognitive evaluation of the importance and personal consequences triggered at the moment that the shocking news is received. This would affect the emotional responses of negative feelings, which, in turn, would lead to the appearance of social behaviors relating to the retelling of the events (rehearsal), thus reinforcing the imprint of the event memory (EM). Therefore, in this model, the direct predictor would be semantic memory, represented in EM.

For the fourth model, Er [5] researched the formation of FBMs in people who experienced the 1999 earthquake in Turkey. She established two groups: one consisting of victims or people with a high degree of involvement and another formed by people who only heard about the earthquake via the media (group of non-victims or with a low degree of involvement). She suggests that the memories of the group with a greater level of involvement are different from those of individuals who did not experience the earthquake directly and that, therefore, two structural models are necessary in order to understand the formation of the FBM. The contribution of Er’s new model [5], in comparison to the previous models, was to find that in subjects who had a direct first-person experience of the event, EM was no different from the memories that those who had only heard about the event had in relation to the context in which they had received the news, thus implying that FBM is equivalent to EM [5].

The models described above propose different relationships between the variables, although some common characteristics can be found in the formation of an FBM: the reaction of shock is always involved; there is a cognitive evaluation of is importance/consequence for the individual; the intensity of emotional reactions plays an essential role; and the aspect of behaviors of open (conversations) and concealed (thoughts) retelling have a direct or indirect influence on the FBM [2,6,9,10].

The tools employed to measure the variables involved in the formation and maintenance of FBMs have also been varied. The first was created by Brown and Kulik [1]. These authors, taking the question *Do you remember the circumstances when you heard for the first time that…?* as a starting point, carried out research on memories of the reception context of nine historical events (among them, the assassinations of J. F. Kennedy, Martin Luther King, and Malcolm X, the attempted assassination of Ronald Reagan, and the death of Francisco Franco in Spain). The results obtained revealed that 50% of the participants coincided in six types of information: where they were when they heard the news; what they were doing; who told them; the emotions shown by those they were with; their own emotions; and the consequences of the event. Furthermore, the participants were asked to evaluate, on a scale of 1 to 5, the consequences of these events with regard to their own lives and to estimate the number of times they had spoken of the event in question. The latter aspect had the aim of establishing the role played by repetition as a determining factor in the formation of an FBM.

The model of evaluation proposed by Brown and Kulik [1] has been widely used, albeit with certain variations. For instance, in 1994, Conway et al. [4] adapted the model for memories regarding the circumstances surrounding hearing news of Margaret Thatcher’s resignation as Prime Minister of the United Kingdom. In their questionnaire, the authors made a differentiation between the attributes of the FBM and encoding mechanisms (levels of surprise, intensity of the surprise, the valence and name of the emotion experienced, among others) and retention (the frequency with which the event was spoken/thought about and with which information was sought in the media). Although no data were provided as far as the reliability or validity of the questionnaire is concerned, the authors proposed a structural equation model to explain the formation of the FBM based on the data obtained with this tool.

Later, in 1998, Finkenauer et al. [6] created a third questionnaire consisting of eight sets of questions to measure the factors involved in the formation and retention of FBMs with regard to the death of King Baudouin of Belgium. The first section evaluated five standard categories of FBMs: exact day and time when the news was heard; the place; people who were present; what people were doing; and up to five specific details of the moment when the news was heard. The second section evaluated the memories of the event in itself, i.e., event memory. The participants answered questions regarding the day, date, and time of death, the place and cause of King Baudouin’s death, and who the first person was to give their condolences to the Queen of Belgium. The third section of the questionnaire measured the overt rehearsal of the news via three items, including information regarding the coverage of the event in the media, general conversations about the event, and conversations focusing on specific details of the event. The questionnaire also included the evaluation of participants’ emotional states (how much the news affected them, how emotionally shocked they were, and how moved they felt), level of surprise (how surprised or shocked they were by the news), and the degree of novelty (how usual or unusual the death of the King was for them and whether they considered that it was a common or uncommon event). Lastly, the importance and personal consequences attributed to the event were evaluated, along with the affective attitudes of the participants toward the royal family (whether they like the royal family or not).

The questionnaire created by Finkenauer et al. [6] complements those put forward by Brown and Kulik [1] and Conway et al. [4], as it addresses the different variables involved in the formation and maintenance of the FBM and, although (as is the case of Conway et al. [4]) it does not offer a detailed analysis of the properties of the tool, the structural equation models carried out based on the observable variables (those collected in the test) make it possible to conclude that their model achieves more acceptable indices for the explanation of the formation and maintenance of the FBM of the death of King Baudouin compared to those of Brown and Kulik [1] and Conway et al. [4].

Later, Er [5] revisited the proposal of Finkenauer et al. [6] and suggested a new model for the formation of FBMs based on memories of the context of receiving the news of the Marmara earthquake, which affected Istanbul, Kocaeli, and other regions of Turkey in 1999. In accordance with the results of the exploratory factor analysis, the tool consisted of 32 items, although there is no description of the eliminated items or the final make-up of the factors. Neither is information provided regarding the internal consistency of each subdimension. However, the confirmatory factor analysis carried out via the structural equation models demonstrates satisfactory indices.

As can be inferred from all of the above, the study of FBMs has traditionally been based on the conjunction of cognitive and social theories that are reflected in the tools used, all of which have been created for the evaluation of the memory of significant public events [11,12,13,14]. However, contemporary research has focused on private or personal events [15], ranging from the mere onset of a menstrual cycle [16], or an invitation to a desired social group [17], to receiving news about something that is personally relevant, such as the death, illness, or accident of a loved one [1,15,18,19].

In this context, mention should be made of studies in which the process of the communication of a diagnosis has been analyzed [20,21,22,23,24,25,26,27,28,29,30]. In these studies, in spite of the li-mitations implied by working with small samples, the results obtained are particularly revealing as they have enabled the identification of numerous deficiencies when bad news is communicated [22]. These include employing nuclear messages [26,30], informing only one parent in a situation of affective vulnerability [27,31], informing the parents before they see the child for the first time [21,28,31], neglecting the necessary conditions of privacy, and giving an excessive amount of negative information [20,24,25,28,29,30], among others. In addition, different studies have shown that the way in which the news of a diagnosis is given can have an influence on how the event is remembered [32,33,34,35].

Implicit within the diagnosis of a disability are several aspects that make it susceptible to the triggering of an FBM: the diagnosis is usually unexpected; it causes surprise; and implies a significant shock for those receiving it. Indeed, the news is not only received as something extremely negative, but a great deal of uncertainty is also generated in the parents regarding the future of their child, among other issues. As a consequence, different (and often ambivalent) cognitive, emotional, and behavioral reactions are triggered in the parents, transforming the diagnosis into a springboard for doubts, insecurities, sorrow, and/or disorientation. Patients (and their family members) who receive an unexpected medical diagnosis often say that they will never forget the moment, that it will stay with them for years, clearly remembering the circumstances of the moment in which the diagnosis was given [36].

Different studies have concluded that the moment in which news is received regar-ding a child’s disability (or risk of suffering one) is engraved in people’s memory, with the precise details of the circumstances being clearly remembered even years after the diagnosis [10,30,31,36,37,38]. However, although the news of the diagnosis in itself may be equally surprising and have similar consequences for the patients, there are many studies that conclude that the way in which the news of the diagnosis is given may have an influence on how the event is remembered [32,33,34,35].

The diagnosis of a child’s visual impairment is a private or personal event in which the parents are directly involved (given their physical presence in the place of the event, as protagonists or as direct witnesses) and maintain a close involvement with the event (as recipients of a significant piece of news for the child and for their lives). However, this circumstance has not received much attention in terms of research. Therefore, several research questions arise, among which the following stand out: Does receiving a diagnosis of a child’s visual impairment generate an FBM for parents? If so, does this FBM depend on the circumstances surrounding the communication of the news? Is it affected by sociodemographic factors and/or the child’s clinical characteristics?

Taking the above into account and understanding by first news the moment in which a professional informs the parents that their child has a disability, disorder, or developmental delay [31], the present study has the objective of discovering how the first news of visual impairment comes about and whether the memory thereof persists over time, generating an FBM.

## 2. Method

### 2.1. Participants

A longitudinal study with an ex post facto retrospective design was carried out in two phases. The criterion employed for the selection of the sample was the fact of being the mother of, at least, one child aged between 0 and 18 years affiliated to the Spanish National Organization of the Blind (ONCE in its Spanish acronym,) in the Autonomous Community of Galicia (Spain). Thus, all mothers who participated in the study had the common factor of being biological mothers of a child with visual impairment, either low vision or blindness.

It should be taken into account that the World Health Organization (WHO) [39] and ONCE [40] classify visual impairment differently. At the same time, the WHO considers a person to be visually impaired if he/she has a visual acuity equal to or less than 0.3 (VA ≥ 0.5 logMAR) with the best possible correction or a visual field of 20 degrees or less to be eligible to join ONCE the person must have at least one of the following visual requirements in both eyes: visual acuity equal to or less than 0.1 (VA ≥ 1.0 logMAR), obtained with the best possible optical correction and/or a visual field reduced to 10 degrees or less.

A total of 38 women aged between 20 and 40 years (*M* = 34.63; *SD* = 4.26) participated in Phase A (2009). The ages of their children ranged from 1 to 18 years (*M* = 8.95; *SD* = 4.81; 52.6% male and 47.4% female). Most mothers were married (89.5%), with an intermediate level of education (39.5% had graduated from high school and 36.8% had studied baccalaureate or vocational training). Their socioeconomic status was middle (55.3%) or low (21.1%). As far as their working situation was concerned, 52.6% were working and 44.8% were unemployed. In Phase B (carried out in 2019), the mean age of the mothers was 45.07 years (*SD* = 3.89), with a range of between 30 and 50. The age range of their children in this phase oscillated between 10 and 28 years of age (*M* = 20.68; *SD* = 4.41). No significant differences were found in the sociodemographic characteristics of the participants between the two phases (see Table 1).

None of the parents (neither mothers nor fathers) were visually impaired. In addition, all children lived with the mother and/or father and none lived with a legal guardian.

### 2.2. Measures

All mothers completed three questionnaires designed ad hoc for the research:-Questionnaire on sociodemographic data, including: age, marital status, level of education, socioeconomic status, employment situation, place of residence, and child’s sex and age.-Questionnaire on child’s clinical variables, including: origin of visual impairment, type of disability, degree of disability, existence (or lack thereof) of other associated disabilities and time since the diagnosis of the visual impairment.-Questionnaire on aspects related to the communication of the diagnosis of visual impairment. This instrument was created in order to discover how the first news of the child’s visual impairment was given and includes: (i) five open-response questions: Who gave the diagnosis? What were you told in the diagnosis? How were you told? When was the diagnosis given? and Where was the diagnosis given? (ii) one closed-ended question (Who was the diagnosis given to?) with three options (to the father, to the mother, to both parents at the same time); and (iii) two dichotomous questions (Yes/No): Would you like to have been told in a different way? and Did you seek confirmation of the diagnosis from other specialists?-Memory index: an ordinal scale created to evaluate the degree of coincidence between the answers given to the Questionnaire on aspects related to the communication of the diagnosis of visual impairment at the two times of data collection (Phase A (2009) and Phase B (2019)). This Likert-type scale reflects the answers given to each of the questions asked in the aforementioned questionnaire (Who gave the diagnosis? What were you told in the diagnosis? How were you told? When was the diagnosis given? Where was the diagnosis given? and Who was the diagnosis given to?). These data were evaluated with three possible options: 0 (=no coincidence); 1 (=partial coincidence); and 2 (=total coincidence). Partial coincidence refers to those cases in which the expression changes (different words are used) but the meaning of the answer is maintained (e.g., if the answer to the question What were you told in the diagnosis? was “that there was no possibility of recovering his/her sight” in Phase A and “he/she would never see again” in Phase B). Total coincidence implies that the expression is maintained in a literal way (e.g., if the answer to the question How were you told? was “with little tact” in both Phase A and Phase B of the research). The score obtained was situated on a scale of 0 (lack of coincidence) to 12 (maximum coincidence) points, providing information on the specificity of the memory (the probability that the memory will be the same regardless of the passing of time). A score close to 12 indicates a greater memory related to the circumstances in which the first news of the diagnosis was given while, on the other hand, a score closer to 0 implies that the memory is less.

It should be highlighted that, when carrying out this research, existing scoring scales (e.g., the Autobiographical Memory Questionnaire (AMQ) [41], the Weighted Attribution Scale [42] or the Flashbulb Memory Checklist (FBMC) [19]) were not employed). This was due to the fact that what was desired was to obtain a record of specific facts in response to the question *How was the first news of your child’s visual impairment given?* in order to then compare personal memories of the event based on the idea that those receiving bad news never forget where, when, and how it was given [1,4,6,10,13,31,43].

### 2.3. Procedure

In order to guarantee alignment to the study object and the comprehension of the questions, all of the questionnaires were submitted for evaluation by an external committee made up of two groups: one of experts with doctorates in different fields (medicine, psychology, and education), who supervised the scientific and ethical issues, and another of mothers with different levels of education (no studies, high school studies, university studies), who reviewed the comprehension of the questions. Following this review, the wording of certain items was modified, and the definitive versions of the questionnaires were obtained.

For the memory index, the following steps were followed: (a) the answers to the six questions relating to the communication of the first news of the diagnosis (who gave the diagnosis, when it was given, to whom, where, what was said, and how it was said) were extracted/collected from the two phases of the research; (b) an ordinal scale was created to evaluate the degree of coincidence between the answers given; (c) two coders, blinded to the objectives and hypothesis of the study, scored each narrative; (d) the Kappa agreement index between the coders was calculated, defined by Landis and Koch [44] as: poor (= 0.00), slight (0.01–0.20), fair (0.21–0.40), moderate (0.41–0.60), substantial (0.61–0.80), and almost perfect (0.81–1.00); and (d) a third coder, also blinded to the objectives and hypothesis, reviewed any discrepancies in the scores and any doubts were resolved consensually via discussion between the original coders.

Given the difficulty of gaining access to such a specific group of people, the selection of the sample was carried out in Phase A via the non-probabilistic snowball sampling technique [45,46]. To this end, members of the ONCE who fulfilled the inclusion criteria established for this study (women with at least one child aged between 0 and 18 who were members of ONCE in the Autonomous Community of Galicia) were contacted and asked to participate. In turn, they were asked to help gain access to other people who could take part in the research. In this way, contact was established with a total of 171 mothers, representing 69.2% of the total target population as, in 2009, the ONCE in Galicia had a total of 247 child members aged from 0 to 18 years of age. Of these mothers, 103 agreed to collaborate with the research, providing a name and address for the documentation to be sent. Three months after the initial mailing, the rate of response was 79.6% with the final sample consisting of 82 subjects representing, approximately, a third of the total population. This can be considered to be adequate and sufficient in order to be able to establish generalizations and to draw conclusions [47].

In the initial contact, the aims of the research were explained, and instructions were given for filling in the questionnaires. It was stressed that participation was voluntary, that anonymity would be guaranteed and that withdrawal from the research was possible at any time. The people who agreed to participate were sent the documentation via postal mail along with a postage-paid response envelope so that their participation would not imply any financial cost for them. The documentation sent included: (a) a consent form taking into account the ethical principles of psychologists and the code of conduct of the American Psychological Association [48]; and (b) a note stating that if a mother wished to receive the results and/or was willing to collaborate in future research, she should send her contact details along with the required documentation (a total of 41 mothers provided this information). No participant was forced to do anything against her will and no identification data were collected (with the exception of those provided by the mothers who expressed a desire to collaborate in future research).

The data collection for Phase B was carried out via e-mail with the 41 mothers who had expressed their willingness to collaborate in future research in 2009. The reasons for contacting them again were explained and they were asked to fill in the attached questionnaires. A link was sent for an online questionnaire, which included the variables studied in 2009, and the data protection protocol was explained. Once the mothers had started the questionnaire, they had the option of saving a partially completed response and continuing at a later time. This option was given in order to enable the mothers to provide more detailed answers to the open-ended questions and to adapt to the needs of those with busy schedules, thus increasing the rates of participation and completion. Only one reply was permitted for each IP address. Two months after the initial e-mail contact, a total of 38 replies had been received (a cooperation rate of 92.68%).

Thus, the final study sample consisted of 38 mothers who collaborated in both phases of the research.

### 2.4. Ethical Considerations

The research process guaranteed the confidentiality of the participants’ personal data at all times in accordance with current Spanish legislation (Organic Law 15/1999) of December 13, on Personal Data Protection [49], which was in force during Phase A of the research, and Organic Law 3/2018 of December 5, on Personal Data Protection and Guarantee of Digital Rights [50], in force during Phase B. Therefore, it can be stated that the principles guiding ethical practice were followed [51]. All participants (in Phase A and Phase B) gave their informed consent for inclusion before they participated in the study. The consent form outlined the confidentiality and anonymity of the participants’ data (to safeguard the participants’ confidentiality, their real names were replaced with numbers to identify them: Mother 1 to Mother 38), the aim and procedure of the study and the option of participating in the study or withdrawing at any moment, along with details of how to contact the research team in order to clarify any doubts. In addition, a password-protected folder was used to store the data, ensuring that no unauthorized person could obtain access to the data and it was guaranteed that they would only be used for the purposes of the research project.

### 2.5. Data Analysis

In order to understand more about the context in which the first news item occurs and to better understand the meaning of the events related to this moment, a mixed methods data analysis was conducted. Thus, the information obtained from the answers to the questions: *Who gives the diagnosis? What were they told in the diagnosis? How were they told*? *When was the diagnosis communicated?* and *Where was the diagnosis communicated?* was transcribed by the research team and then qualitatively analyzed using an inductive system of categories and codes created to make sense of the information collected. To facilitate information processing, the qualitative software MaxQDA 12 was employed to analyze the data.

The data analysis was carried out in two phases. In the first phase of coding, we used a broad and generic category system. In the second phase, new subcodes for the fundamental themes and ideas emerged (Table 2).

Finally, we analyzed each of these subcodes in depth to see if they could be broken down further or merged with other codes.

All information was categorized, discussed, and analyzed by the research team, organizing the doubtful information.

In addition, a descriptive analysis was carried out of the sociodemographic and clinical variables considered, including mean and standard deviation for the quantitative variables and frequency and percentage in the relevant categories. In order to analyze differences between the groups, the χ^2^ test was applied for cross-tabulation, and Cohen’s Kappa index was used as an agreement test between phases for the variables referring to the context and content of the communication of impairment. All of these data were analyzed with IBM SPSS 27.0 for Windows.

## 3. Results

### 3.1. Time since Diagnosis and Clinical Profile

The period of time elapsed since the mothers first received the diagnosis of their children’s visual impairment in Phase A was, in most cases, between 5 and 10 years (*M* = 7.57; *SD* = 4.55), and in Phase B between 15 and 20 years (*M* = 18.00; *SD* = 4.71). In both phases, most children presented low vision (73.7%), of congenital origin (71.1%), with a profound degree of disability greater than 75% (73.6%), and with no other disabilities associated with visual impairment (63.2%) (see Table 3).

The possible existence of significant differences in types of visual impairment were analyzed between the two phases but as expected, none were found.

### 3.2. Circumstances in Which the “First News” Was Given: Who, When, to Whom, Where, What, and How the Diagnosis Was Given

The circumstances in which the first news of the visual impairment was given are outlined in Table 4.

As far as the question of who the professional was that gave the diagnosis is concerned, in both phases of the research, the majority of the mothers (47.4%) stated that it was an ophthalmologist. As for when they were notified of the diagnosis, the greatest percentage (31.6%) was situated in the interval of “six months to a year” of the child’s life.

In general, both parents were present when the diagnosis was given (60.5%). However, in a high percentage of cases (29.0%), the diagnosis was given only to the mother, leaving her with the responsibility of giving the news to her partner:

“*After telling me, the doctor left the room. I was in a state of shock, trying to take in the news and thinking of how to tell my husband”*.(Mother 2)

*“I think it would have been better if the diagnosis had been given to us both together”*.(Mother 16)

*“When I saw my husband, all I could do was to cry and cry. I didn’t know how to tell him what I had been told”*.(Mother 6, Mother 20, Mother 33, and Mother 37)

Concerning the place in which the first news of the diagnosis was given, more than 70% of the mothers said they had been told in the office of the professional responsible for transmitting the news. It should be highlighted, however, that, in other cases, the news was given in other, less appropriate, places, such as in the hospital ward (18.4%), a corridor or waiting room (2.6%), and even in the delivery room (2.6%). The place in which the diagnosis is given is clearly important for the mothers, as is reflected in the following comment:

*“We were given the news in the hospital room, which we shared with other parents. We would like to have been told alone. It was difficult for us and, I think, for the other parents too as they were happy with their babies and did not know how to act with us after the doctor had left. I think it is an extremely delicate moment which requires a certain degree of privacy”*.(Mother 13)

The type of information given was, for a high percentage of mothers (76.3%), of a medical nature:

*“Atrophy of the optical nerve”*.(Mother 2)

*“Glioma”*.(Mother 3)

*“PEHO syndrome”*.(Mother 6)

*“Periventricular leukomalacia and cortical blindness*”.(Mother 9)

*“Suffering from a pale optic nerve”*.(Mother 12)

*“Congenital cataracts*”.(Mother 19)

*“Congenital glaucoma”*.(Mother 28)

*“Massive cerebral infarction due to strangulation with the umbilical cord”*.(Mother 29)

*“Bilateral amaurosis”*.(Mother 32)

Etc.

whereas 13.2% defined it, in Phase A, as “*unspecific*” or as “*abstract*” and “*imprecise*” in Phase B:

*“They didn’t know exactly what the diagnosis was because the child could not collaborate”*.(Mother 33)

*“That nothing good was going to happen”*.(Mother 23)

*“In the first diagnosis, they didn’t know what it was, but it looked bad”*.(Mother 31)

*“That as the baby was born prematurely, there could be many consequences”*.(Mother 18 and Mother 35)

According to the mothers, the information given in the diagnosis makes it difficult at times to understand what is happening, as reflected in the following comments:

“*The language used was so technical that I couldn’t really understand what they were telling me or what it implied*”.(Mother 1, Mother 2, Mother 6, Mother 7, Mother 9, Mother 12, Mother 15, Mother 27, Mother 32, Mother 34, Mother 36, and Mother 37)

“*After receiving the news, I didn’t know what the real situation of my daughter was, if she could see or not, how much she could see…*”.(Mother 23, Mother 31, and Mother 35)

In addition to analyzing the type of information that is given, it is necessary to know *how* it is transmitted. Almost half of the participants in both phases (44.7%) highlighted the lack of sensitivity of the professionals when giving the news (“*with little tact*”—Mother 6, Mother 8, Mother 10, Mother 13, Mother 15, Mother 16, Mother 17, Mother 24, Mother 25, Mother 26, Mother 30, Mother 32, Mother 34, Mother 35, Mother 36, Mother 37, and Mother 38).

The χ^2^ test for cross-tabulation was applied to analyze the existence of any significant differences between the circumstances in which the diagnosis was given (to whom it was given, where, who gave it, and when it was given) and the clinical profile (congenital/acquired disability and presence (or lack thereof) of other associated impairments). A significant association was found between the origin/cause of the visual impairment (acquired vs. congenital) and to whom the diagnosis was given (father/mother), with the news being given to the mother alone in 100% of the cases in which the impairment was acquired. Likewise, the origin of the impairment also introduced significant differences with regard to the place in which the news was given. In contrast, in 90.9% of the acquired cases, the news was given in the doctor’s office, this percentage decreased to 70.3% in cases of congenital impairment, which was communicated in the delivery room or on the hospital ward in a considerable percentage of cases (see Table 5). The presence of other associated impairments showed statistically significant differences in relation to who gave the diagnosis and when and where it occurred. Thus, when there were no other associated disabilities: the diagnosis of visual impairment was given, on the whole, by an ophthalmologist; the diagnosis was given between the first and fourth month of life; the diagnosis was given in the doctor’s/specialist’s office. On the other hand, when there were other associated disabilities present: other professionals (not an ophthalmologist) played a key role in giving the first news of the diagnosis; the diagnosis was mainly given between 6 and 12 months of age; in a large percentage of cases, the diagnosis was given in the hospital ward (see Table 5).

With the aim of evaluating the degree of satisfaction with the diagnosis, the mothers were asked whether they would have liked to have received the diagnosis in a different way and, also whether, after receiving the first news, they had sought a second opinion. A total of 44.7% of the participants expressed a desire to have received the news in a different way, and a high percentage of them (71.1%) sought confirmation of the diagnosis from another specialist (see Table 6).

After receiving news of the disability, a high percentage of mothers (71.1%) admitted to having sought a second opinion, with many of them consulting more than two specialists in order to confirm the initial diagnosis (70.4%). No significant relationship has been found between this seeking of a second opinion and the type of visual impairment (blindness/partial sight), the origin thereof (congenital/acquired), or the presence of other associated disabilities. Neither were there any statistically significant differences relating to this variable and the circumstances in which the diagnosis was given (what, how, when, and to whom the news was given).

As far as the degree of satisfaction with the way in which the diagnosis was given is concerned, there is a notably positive and statistically significant association between this variable and the content of a message conveying an irreversible diagnosis (“*There is no possibility of the child recovering his/her sight*”) (χ^2^ (2) = 5.530, *p* = 0.030) and a diagnosis given “*with little tact*” (χ^2^(3) = 18.704, *p* < 0.001). In both cases, the desire was expressed for the news to have been given in a different way (see Table 7).

### 3.3. Flashbulb Memories and Conditioning Variables

The answers given by the mothers who participated in Phase B were compared with those that each of them had given in Phase A. The results obtained by the coders who evaluated each narrative, with 92.1% coincidence in their responses and an almost perfect Kappa agreement index (*k* = 0.843; *p* < 0.001), confirm that the memory of the first news of the diagnosis is, indeed, long-lasting.

The information obtained regarding ***how*** the news was given is particularly worthy of note, as when doctors gave the news “*with little tact*”, the response was identical in 100% of the cases in both phases of the research (see Table 4), obtaining significant differences (χ^2^ (6) = 76.000; *p* < 0.001), with regard to other ways of giving the news.

Furthermore, the mean obtained in the memory index was extremely high, indicating little variability or dispersion in the answers given (*M* = 11.44, *SD* = 0.50). An analysis of the data offered by this index demonstrates that 10 years after the first evaluation, 53.3% of the mothers remembered, in a literal way, the way in which they had received the diagnosis of their child’s visual impairment and that for the remaining 44.7% there was a partial coincidence between the answers given, as the meaning was maintained even though the same words were not used. These results coincide with the subjective perception that the participants have regarding their memory of the first news of the diagnosis, confirming the idea that the impact thereof is so strong that it endures in the memory. Indeed, when in 2019 (Phase B), they were asked whether they believed that the passing of time had affected their memories regarding the diagnosis, and the answer was negative in 100% of the cases.

Therefore, the flashbulb memory is confirmed, as is the fact that its formation does not depend on sociodemographic factors or on the clinics included in the study, given that, taking the memory index as a dependent variable, it was found that none of the variables analyzed introduce statistically significant differences.

## 4. Discussion

The aim of the present study was to consider how the first news of a child’s visual impairment is given and whether the memory of this personal event persists with the passing of time, generating a flashbulb memory.

Most cases in the study concern children presenting low vision. This clinical profile coincides with the data published by the ONCE [52], as only 8% of its members aged 0–5 and 9% aged 6–16 are classified as blind, with a clear tendency for new members of any age to have partial vision.

The absolute coincidence of the answers provided in the two phases of the research demonstrates, as is the case in other studies [4,6,30,31,36,37,43], that the shock caused by the first news of such a diagnosis in those who receive it (in this case, the mothers), remains intact in their memory, as they remember where and when the diagnosis was given and how they were informed. Although the results of this study represent an extremely specific population (mothers of visually impaired children), they confirm that the impact of receiving the first news of a child’s diagnosis gives rise to a vivid and intense memory of the details surrounding it, thus coinciding with the conclusions of other authors [1,4,6,13,36,43,53,54,55,56].

In more than 75% of the cases, the diagnosis was given at the time of birth or during the first year of life, as is the case in other studies [57,58], with an ophthalmologist being responsible for transmitting the diagnosis in most cases. However, when other disabilities were present in addition to visual impairment, the diagnosis was made earlier, generally at the time of the child’s birth or in the first week of life or, as Pérez [59] states, between the age of two and three months, and may imply the intervention of other specialists.

In general, the first news of the diagnosis was given to both parents at the same time. However, in a large percentage of cases, the mothers were given the diagnosis alone, as also occurred in López Montellano’s study [58]. This was particularly the case when the child’s visual impairment was acquired. This may be explained by gender roles due to the fact that, as stated by Rolland [60] and Tates and Meeuwesen [61], in cases of children with disabilities and/or chronic illnesses, on the whole, the father takes on the role of breadwinner for the family while the mother becomes an informal carer. In such cases, as can be deduced from the mothers’ responses, being responsible for transmitting the news to their partners causes a great emotional burden, which is added to the shock caused by the diagnosis itself. On the other hand, when the visual impairment was congenital, a significant percentage of fathers received the news alone, which could be due to the fact that, as stated by Ponte et al. [31], after the birth, the father is the person whom professionals address to provide information in relation to how the mother and/or baby are.

The place in which the news is given (in general, the doctor’s office) can be considered to be appropriate. As stated by different authors [20,25,28,29,30,54,62,63,64], this is a private space that makes it possible to create a climate of trust and helps to maintain confidentiality. This is not the case with other places mentioned, such as the delivery room, the hospital ward or a corridor, or any other available room, as is also mentioned in different studies [20,25,28,29,30,37,63,65,66,67]. The choice of these places for giving the news may be motivated by the desire to provide information about the disability from the moment in which the diagnosis is established due to the fact that a delay would only cause more anguish and uncertainty and would lead to a deterioration in the relationship with, and trust in, the doctor [37,68]. However, such practices should be reconsidered if the comments of the mothers obtained in the present study are to be taken into account.

The answers given by the mothers with regard to the type of content and how this was transmitted by the doctor in relation to the diagnosis indicate that it was a “medical diagnosis” or consisted of “unspecified information” given “with little tact”. In general, it did not help them to understand the scope of their children’s visual impairment, which is worthy of note as if the first news is given without the use of clear terminology or an argument that helps the family to understand the significance of their child’s disability, a great deal of insecurity may be caused [69,70,71]. The lack of information, along with an unclear and incomprehensible diagnosis, may lead to families seeking the opinion of a series of other doctors who they hope may offer them a more favorable perspective [58,71,72,73,74], thus delaying intervention. This is confirmed in the results of the present study, as more than 71% of the mothers stated that they had sought a second medical opinion to provide a greater degree of certainty and diagnostic confirmation. Indeed, more than 70% admitted to having consulted more than two specialists in order to confirm the initial diagnosis, independently of the type of disability and the presence of other associated disabilities.

## 5. Conclusions

The main finding of this research is the confirmation that the first news of a diagnosis produces long-lasting memories, thus demonstrating that memories of a medical diagnosis remain vivid, as is the case with memories of public events. More significantly, the results show that, although the development and persistence of such memories do not depend on the sociodemographic or clinical factors included in this study, the way in which the professional gives the first news (how it is said) does appear to play a significant role in the memory generated. As a consequence, the results obtained are of great interest and suggest that improvements are necessary in medical practice in the communication of the first news of a disability. In this regard, it is strongly recommended that the person/people receiving the diagnosis should be given the time they need to assimilate the news, that no more information should be given than is required [37,68,75,76,77,78], that the news should be transmitted clearly, with no ambiguity or technical, medical language that may lead to greater uncertainty in those receiving the message. In other words, as Lillo [66] states, it is necessary to listen and wait for the most appropriate time to intervene, to verify that the information given has been understood, and to address any possible doubts, fears, and/or concerns that may arise upon receiving the news. Empathy must be shown and trust transmitted. Furthermore, the information must be given to both parents at the same time or, at the least, in the presence of another professional, such as a psychologist, who can offer emotional support for a diagnosis that is usually unexpected. Finally, the information should be given in a quiet, comfortable, and private place in order to facilitate a suitable relationship of communication between the doctor and those receiving the news.

The results obtained in this study have significant implications in clinical practice as they make it clear that there is a need to train medical professionals with regard to appropriate procedures for the communication of bad news. Such training would, therefore, lead to the memory of receiving the initial diagnosis being positive (assistance, support, empathy, etc.) rather than negative (lack of information about the diagnosis, lack of understanding from healthcare professionals, fear, anxiety, etc.).

More research is needed to complement the data obtained in this study with regard to the development of such memories in relation to the circumstances surrounding the reception of the news, the emotional impact of the diagnosis received, and/or the consequences of the diagnosis on the life of the individual, among other factors.

## 6. Limitations and Future Work

One aspect that may limit the results of the present study can be observed in the sample recruited via non-probabilistic snowball sampling. This may have led to the fact that only mothers who were most affected by the initial news of their child’s diagnosis participated in the study, which, according to Brown and Kulik [1], would make them prone to generating an FBM after the event. Another limitation can be found in the fact that neither the mothers’ prior knowledge of visual impairment nor their expectations for their children before the diagnosis were taken into account.

With regard to future research, it is necessary, as already stated by different authors [13,53,54,55], to understand which factors contribute to the development of flashbulb memories (the circumstances surrounding the reception of the news, the emotional impact of the diagnosis received, everyday contact with the consequences of the news (the disability) in the daily life of the subject who received the diagnosis and/or the consequences of the diagnosis in the life of the individual, among other factors). We also consider it necessary to include the impact of such memories on children’s development.

## Figures and Tables

**Table 1 children-10-00881-t001:** Sociodemographic characteristics of the sample.

	Phase A (*n* = 38)		Phase B (*n* = 38)		Kappa (*k*)
	*n*	%	*n*	%	
Age range					1.000 ***
20–30 years of age	6	15.8	-	-	
31–40 years of age	32	84.2	6	15.8	
41–50 years of age	-	-	32	84.2	
Marital Status					0.636 ***
Single	1	2.6	1	2.6	
Married	34	89.5	33	86.8	
Separated/Divorced	2	5.3	4	10.6	
Cohabiting with a partner	1	2.6	-	-	
Level of Education					1.000 ***
No studies	1	2.6	1	2.6	
High school or similar	15	39.5	15	39.5	
Baccalaureate or vocational training	14	36.8	14	36.8	
University studies	8	21.1	8	21.1	
Socioeconomic Status					0.869 ***
Low (<600 EUR/month)	5	13.2	4	10.5	
Medium–Low (600–1000 EUR/month)	8	21.1	7	18.5	
Medium (1000–1500 EUR/month)	21	55.3	23	60.5	
Medium–High (1500–2000 EUR/month)	4	10.5	4	10.5	
High (>2000 EUR/month)	-	-	-	-	
Place of residence					1.000 ***
Rural (<5000 inhabitants)	10	26.3	10	26.3	
Semi-urban (5000–50,000 inhabitants)	12	31.6	12	31.6	
Urban (>50,000 inhabitants)	16	42.1	16	42.1	
Employment situation					0.791 ***
Working	20	52.6	24	63.2	
Unemployed	17	44.8	14	36.8	
Leave of absence	1	2.6	-	-	
Child’s sex					1.000 ***
Male	20	52.6	20	52.6	
Female	18	47.4	18	47.4	
Child’s age range					1.000 ***
0–3 years of age/10–13 years of age	4	10.5	4	10.5	
4–6 years of age/14–16 years of age	11	28.9	11	28.9	
7–10 years of age/17–20 years of age	7	18.4	7	18.4	
11–14 years of age/21–24 years of age	11	28.9	11	28.9	
15–18 years of age/25–28 years of age	5	13.2	5	13.2	

*** *p* < 0.001.

**Table 2 children-10-00881-t002:** Categories and codes system.

Categories	Codes
Who gave the diagnosis?	1. The ophthalmologist. 2. The pediatrician. 3. Other medical professionals (neurologist, neurosurgeon, oncologist, gynecologist, intensive care professional in the premature unit).
What did they tell you at diagnosis?	1. A medical diagnosis/What the child had in medical terms. 2. Unspecified information/Abstract information, vague, unspecific. 3. That there was no possibility of recuperating their vision/That he/she would never see again.
How was it told you?	1. In a straightforward manner, using medical terms/With medical terminology. 2. Speaking naturally/With quite a lot of empathy. 3. Gently and tactfully/With quite a lot of empathy. 4. With little tact/With little tact.
When did you receive the diagnosis?	1. From the moment of birth and the first week. 2. Between one and four months old. 3. Between six months and one year old. 4. Between two and six years old.
Where was the diagnosis communicated?	1. In the delivery room. 2. In the hospital bedroom. 3. In the doctor’s (or other professional’s) office. 4. In the corridor/ In any available room.

**Table 3 children-10-00881-t003:** Characteristics of the visual impairment and the time since the diagnosis.

	% Phase A (*n* = 38)		% Phase B (*n* = 38)		Kappa (*k*)
	*n*	%	*n*	%	
Origin of visual impairment					1.000 ***
Congenital/Visually impaired at birth	27	71.1	21	71.1	
Acquired/After birth	11	28.9	11	28.9	
Type of visual impairment					1.000 ***
^a^ Low vision	28	73.7	28	73.7	
^b^ Blindness	10	26.3	10	26.3	
Degree of disability					1.000 ***
Moderate (25–49%)	5	13.2	5	13.2	
Severe (50–74%)	5	13.2	5	13.2	
Profound (>75%)	28	73.6	28	73.6	
Presence of other impairments in addition to visual impairment					1.000 ***
Yes	14	36.8	14	36.8	
No	24	63.2	24	63.2	
Time since the diagnosis					1.000 ***
0–2 years/10–12 years	5	13.2	5	13.2	
3–4 years/13–14 years	6	15.8	6	15.8	
5–7 years/15–17 years	8	21.1	8	21.1	
8–10 years/18–20 years	9	23.7	9	23.7	
11–14 years/21–24 years	6	15.8	6	15.8	
>14 years/>24 years	4	10.5	4	10.5	

^a^ He/she still has some useful vision for his/her daily activities (with the best possible correction, he/she can see or distinguish, albeit with great difficulty, some objects at close distance). ^b^ He/She cannot see at all or only has a vague perception of light (perhaps he/she can distinguish between light and darkness but not the shape of objects). *** *p* < 0.001.

**Table 4 children-10-00881-t004:** Circumstances in which the first news was given.

	Phase A (*n* = 38)		Phase B (*n* = 38)		Kappa (*k*)
	*n*	%	*n*	%	
Who gave the diagnosis?					1.000 ***
An ophthalmologist	18	47.4	18	47.4	
A pediatrician	8	21.1	8	21.1	
Other medical professionals (neurologist, neurosurgeon, oncologist, gynecologist, intensive care professional in the premature unit)	12	31.5	12	31.5	
When was the diagnosis given?					1.000 ***
At the moment of birth or during the first week	6	15.8	6	15.8	
Between one and four months old	10	26.3	10	26.3	
Between six months and one year old	12	31.6	12	31.6	
Between two and six years old	10	26.3	10	26.3	
To whom was the diagnosis given?					1.000 ***
The father	4	10.5	4	10.5	
The mother	11	29.0	11	29.0	
Both at the same time	23	60.5	23	60.5	
Where was the diagnosis given?					1.000 ***
In the delivery room	1	2.6	1	2.6	
On the hospital ward	7	18.4	7	18.4	
In the doctor’s (or another professional’s) office	29	76.4	29	76.4	
In the corridor/In any available room	1	2.6	1	2.6	
What was said in the diagnosis? Phase A/Phase B					1.000 ***
A medical diagnosis/What the child had in medical terms	29	76.3	29	76.3	
Unspecific/abstract/vague information	5	13.2	5	13.2	
That there was no possibility of the child recovering his/her vision/That he/she would never see again	4	10.5	4	10.5	
How the participants were told. Phase A/Phase B					0.697 **
In a straightforward manner, using medical terms/With medical terminology	5	13.1	5	13.1	
Speaking naturally/With quite a lot of empathy	8	21.1	8	42.1	
Gently and tactfully/With quite a lot of empathy	8	21.1	8	21.1	
With little/no tact/With little/no tact	17	44.7	17	44.7	

** *p* < 0.01; *** *p* < 0.001.

**Table 5 children-10-00881-t005:** Circumstances of the diagnosis based on the origin of the disability and the presence of other associated disabilities.

		**Origin of Visual Impairment**				**Total**		**g.l.**	**χ^2^**
		**Congenital**		**Acquired**					
		** *n* **	** *%* **	** *n* **	** *%* **	** *n* **	**%**		
To whom was the diagnosis given?								1	1.364 *
	The father	4	33.3	-	-	4	26.7		
	The mother	8	66.7	3	100.0	11	73.3		
Total		12	100.0	3	100.0	15	100.0		
Where was the diagnosis given?								3	1.978 *
	In the delivery room	1	3.7	-	-	1	2.6		
	On the hospital ward	6	22.3	1	9.1	7	18.4		
	In the doctor’s (or another professional’s) office	19	70.3	10	90.9	29	76.4		
	In the corridor/In any available room	1	3.7	-	-	1	2.6		
Total		27	100.0	11	100.0	38	100.0		
		**Presence of Other Impairments**				**Total**		**g.l.**	**χ^2^**
		**No**		**Yes**					
		** *n* **	** *%* **	** *n* **	** *%* **	** *n* **	**%**		
Who gave the diagnosis?								2	7.738 *
	An ophthalmologist	15	62.5	3	21.4	18	47.4		
	A pediatrician	5	20.8	3	21.4	8	21.1		
	Other medical professionals	4	16.7	8	57.2	12	31.5		
Total		24	100.0	14	100.0	38	100.0		
When was the diagnosis given?								3	10.710 **
	Birth—1st week	2	8.3	4	28.6	6	15.8		
	1–4 months	10	41.7	-	-	10	26.3		
	6–12 months	5	20.8	7	50.0	12	31.5		
	2–5 years old	7	29.2	3	21.4	10	26.3		
Total		24	100.0	14	100.0	38	100.0		
Where was the diagnosis given?								3	11.494 **
	In the delivery room	1	4.2	-	-	1	2.6		
	On the hospital ward	1	4.2	6	42.9	7	18.4		
	In the doctor’s (or another professional’s) office	22	91.6	7	50.0	29	76.3		
	In the corridor/In any available room	-	-	1	7.1	1	2.6		
Total		24	100.0	14	100.0	38	100.0		

* *p* < 0.05; ** *p* < 0.01.

**Table 6 children-10-00881-t006:** Percentage distribution of satisfaction with the diagnosis received and seeking a second opinion.

		Phase A (*n* = 38)		Phase B (*n* = 38)		Kappa (k)
		*n*	%	*n*	%	
Would you have liked to have been told in a different way?						1.000 ***
	Yes	17	44.7	17	44.7	
	No	21	55.3	21	55.3	
	Total	38	100.0	38	100.0	
Did you seek confirmation of the diagnosis from other specialists?						1.000 ***
	Yes	27	71.1	27	71.1	
	No	11	28.9	11	28.9	
	Total	38	100.0	38	100.0	
Number of doctors visited						1.000 **
	1–2	8	29.6	8	29.6	
	>2	19	70.4	19	70.4	
	Total	27	100.0	27	100.0	

** *p* < 0.01; *** *p* < 0.001.

**Table 7 children-10-00881-t007:** Satisfaction with the diagnosis based on how the first news was given.

		Would You Have Liked the News to Have Been Given in a Different Way?				Total		g.l.	χ^2^
		No		Yes					
		*n*	*%*	*n*	*%*	*n*	*%*		
What were the participants told in the diagnosis?								2	5.530 *
	A medical diagnosis was given	18	85.7	11	64.8	29	76.3		
	Vague information was given	3	14.3	2	11.7	5	13.2		
	Told that there was no way the child would see again	-	-	4	23.5	4	10.5		
Total		21	100.0	17	100.0	38	100.0		
How were the participants told?								3	18.704 ***
	In a straightforward manner using medical terminology	4	19.0	1	5.9	5	13.1		
	Gently and tactfully	8	38.1	-	-	8	21.1		
	Speaking naturally	6	28.6	2	11.8	8	21.1		
	With little tact	3	14.3	14	82.3	17	44.7		
Total		21	100.0	17	100.0	38	100.0		

* *p* < 0.05; *** *p* < 0.001.

## Data Availability

The data (anonymized, with no identifying information) are available upon reasonable request to the corresponding author.

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
