# Peer review of "Flashbulb Memories in the Communication of the Diagnosis of Visual Impairment: The Effect of Context and Content"

_children, 2023, doi:10.3390/children10050881_

Round 1

Reviewer 1 Report

This article is on a very important and relevant topic. I have just a few questions/suggestions: 

Methods

Were the mothers included only biological or also adoptive mothers etc.?

Please add the unit of VA - LogMAR? 

Did you ask if the mother/father had a visual impairment? This might affect how the news was received/ impact the news delivery/anticipation if they already had a trusted partnership with the doctor?

you talk about father/mother - what about guardians if the child was not with the father/mother? 

Discussion/conclusions

Did you consider whether medical professionals were trained adequately to deliver this new - may just be a small point to include in discussion/conclusions 

Generally some of the sentences are very long and wordy - they could be made more concise and clearer  

Reviewer 2 Report

The author first described the existing research on Flash bulb memories, and we can see that this section is relatively detailed and has important reference value for the author. Then, the author described the experiment in detail, introducing the situation of the subjects, the experimental process, and the data analysis of the experimental results. The workload seemed very rich. Finally, the author summarizes and prospects the research work. This is a good study, and after some minor revisions, it is recommended to consider publishing it.

The following is a suggestion: It is recommended that the author supplement the introduction of stimuli in the experiment, that is, the specific details of the materials or objects that are being stimulated to the subject.

Reviewer 3 Report

The introduction is quite tedious, It needs to be cut back to a great extent. There are information that is not required indeed. 

It will be interesting if the authors may include the impact of such memories on children's development or milestones if any in the introduction.

The sample size is very small, estimating kappa agreement may not be appropriate. it can be mentioned in the limitation. 
